# The Role of Endogenous Specialized Proresolving Mediators in Mast Cells and Their Involvement in Inflammation and Resolution

**DOI:** 10.3390/ijms26041491

**Published:** 2025-02-11

**Authors:** Nobuyuki Fukuishi, Kentaro Takahama, Hiromasa Kurosaki, Sayaka Ono, Haruka Asai

**Affiliations:** 1Department of Pharmacology, Graduate School of Pharmaceutical Sciences, Kinjo Gakuin University, Nagoya 463-8521, Japan; h-kurosaki@kinjo-u.ac.jp (H.K.); y1971028@kinjo-u.ac.jp (S.O.); asai-h@kinjo-u.ac.jp (H.A.); 2Technology Center, Tokai National Higher Education and Research System, Nagoya 464-8601, Japan; cfa@tech.thers.ac.jp

**Keywords:** mast cells, specialized proresolving mediators, inflammation, arachidonic acid, eicosapentaenoic acid, docosahexaenoic acid, lipoxin, resolvin, protectin, maresin

## Abstract

Many polyunsaturated fatty acids within cells exhibit diverse physiological functions. Particularly, arachidonic acid is the precursor of highly bioactive prostaglandins and leukotrienes, which are pro-inflammatory mediators. However, polyunsaturated fatty acids, such as arachidonic, docosahexaenoic, and eicosapentaenoic acids, can be metabolized into specialized proresolving mediators (SPMs), which have anti-inflammatory properties. Given that pro-inflammatory mediators and SPMs are produced via similar enzymatic pathways, SPMs can play a crucial role in mitigating excessive tissue damage induced by inflammation. Mast cells are immune cells that are widely distributed and strategically positioned at interfaces with the external environment, such as the skin and mucosa. As immune system sentinels, they respond to harmful pathogens and foreign substances. Upon activation, mast cells release various pro-inflammatory mediators, initiating an inflammatory response. Furthermore, these cells secrete factors that promote tissue repair and inhibit inflammation. This dual function positions mast cells as central regulators, balancing between the body’s defense mechanisms and the need to minimize tissue injury. This review investigates the production of SPMs by mast cells and their subsequent effects on these cells. By elucidating the intricate relationship between mast cells and SPMs, this review aims to provide a comprehensive understanding of the mechanism by which these cells regulate the delicate balance between tissue damage and repair at inflammatory sites, ultimately contributing to the resolution of inflammatory responses.

## 1. Introduction

Many very-long-chain fatty acids (VLCFAs) in cell membranes play crucial roles in regulating physiological functions in the human body. Both ω-3 α-linolenic acid and ω-6 linoleic acid, which are VLCFAs, are essential fatty acids that cannot be endogenously synthesized and must be supplied through the diet [1].

α-Linolenic acid is metabolized to eicosatetraenoic acid by fatty acid elongase (ELOVL) 5 and delta-6 desaturase (FADS2) and subsequently converted to eicosapentaenoic acid (EPA) by delta-5 desaturase (FADS1). After conversion from EPA to docosapentaenoic acid (DPA) by ELOVL2/5, DPA undergoes additional elongation by ELOVL2 to form tetracoeicosapentaenoic acid. Subsequently, this intermediate is desaturated by FADS2 and subjected to peroxisomal oxidation to yield docosahexaenoic acid (DHA) (Figure 1) [2]. Linoleic acid undergoes the first metabolic conversion to dihomo-γ-linolenic acid by ELOVL5 and FADS2 and is subsequently converted to arachidonic acid (ARA) by FADS1 [2]. ARA can undergo a series of elongation and desaturation reactions catalyzed by ELOVL2/5 and FADS2, leading to the formation of tetracoeicosapentaenoic acid. Subsequently, this intermediate is converted to DHA by FADS2 and peroxisomal oxidase (Figure 1) [2]. Additionally, ARA is a substrate for cyclooxygenase (COX) and lipoxygenase (LOX), which catalyze the conversion of ARA to various eicosanoids, including prostaglandins (PGs), thromboxanes, and leukotrienes (LTs). Therefore, ARA is a well-established precursor for the synthesis of eicosanoids. Besides ARA, specific VLCFAs, including EPA, DPA, and DHA, can be metabolized by multiple enzymes, such as aspirin-treated COX-2, 5-LOX, 12-LOX, and 15-LOX, giving rise to lipoxins (LXs), resolvins (Rvs), protectin D (PD), and maresins (MaRs) (Figure 2, Figure 3, Figure 4 and Figure 5) [3]. These endogenous lipids have anti-inflammatory properties [4] and are categorized as specialized proresolving mediators (SPMs) [5]. Although the biosynthesis of many pro-inflammatory lipid mediators is completed within a single cell, the generation of anti-inflammatory lipid mediators involves multiple enzymatic steps and intermediate products, thereby requiring a transcellular biosynthesis process that spans several cells [6,7]. COX-2 and 5-LOX are abundantly expressed in various cell types, such as macrophages [8], neutrophils, eosinophils, mast cells, and dendritic cells [9], whereas 12-LOX is predominantly expressed in macrophages [10], platelets, vascular smooth muscle cells, and keratinocytes, and 15-LOX is expressed in macrophages, epithelial cells, vascular endothelial cells, and eosinophils [11,12]. Therefore, SPM biosynthesis is a sequential process involving multiple cell types, each expressing a specific profile of these enzymes.

Mast cells express the high-affinity receptor for the Fc region of immunoglobulin E (FcεRI) on their surface. They are activated by FcεRI cross-linking by antigens to produce PGs and LTs and release various cytokines and histamine, contributing to inflammatory and allergic responses. Based on these findings, mast cells are primarily considered “inflammatory” cells due to their critical role in initiating and perpetuating inflammatory and allergic responses [13]. Conversely, mast cell-derived histamine plays a crucial role in tissue repair and remodeling [14,15]. Additionally, mast cells enhance fibroblast infiltration and proliferation [16], whereas the zinc contained within mast cell granules promotes tissue repair and regeneration [17]. Mast cells have been implicated in the promotion of angiogenesis in tissues [18]. Furthermore, local tissue repair has been reported to be delayed in mast cell-deficient mice (*Kit^W^*/*Kit^W−v^*) [19], indicating that although mast cells are primarily considered “inflammatory” cells, they play a crucial role in resolving inflammation and promoting tissue repair. These findings strongly support the notion that mast cells play a role in regulating the initiation and resolution of inflammatory responses [20,21].

This review aimed to elucidate the role of SPMs in mast cell-controlled local inflammation. Additionally, it aimed to discuss SPM production by mast cells, the effects of SPMs on mast cells, and the general biosynthetic pathways and physiological functions of SPMs, thereby providing a comprehensive overview of the interplay between mast cells and SPMs in local inflammation control.

The metabolism of α-linolenic acid involves a series of elongation and desaturation steps catalyzed by specific elongases and desaturases. This pathway produces eicosapentaenoic acid (EPA) and docosapentaenoic acid (DPA) as intermediates and ultimately culminates in the synthesis of docosahexaenoic acid (DHA) (left panel). Similarly, linoleic acid undergoes elongation and desaturation to form arachidonic acid (ARA) as a key intermediate. Arachidonic acid can then undergo further elongation and desaturation to form DHA, converging on the same end product as the α-linolenic acid pathway (right panel).

## 2. Lipoxins

### 2.1. Biosynthetic Pathways of Lipoxins and Their Relationship with Mast Cells

Bioactive lipids, such as LXs, are characterized by the presence of a trihydroxytetraene moiety. They were initially purified from human leukocytes during the mid-1980s [22]. The generation of LXs involves two or more biosynthetic pathways. The first pathway involves the peroxidation of ARA at the C15 position by 15-LOX in eosinophils and various epithelial cells. The resulting intermediate, 15-hydroperoxyeicosatetraenoic acid (15-HpETE), is taken up by polymorphonuclear cells and monocytes, where it is further metabolized by 5-LOX and epoxide hydrolase to form LXA_4_ and LXB_4_ (Figure 2) [2,23]. Through this pathway, the generation of LXA_4_ and B_4_ in polymorphonuclear leukocytes shunts the activity of 5-LOX toward the synthesis of these LXs. Consequently, the production of LTA_4_ and other metabolites by 5-LOX is reduced [24]. Additionally, the 15-HpETE generated in the first step of this pathway exhibits uptake, particularly by polymorphonuclear leukocytes. This uptake stimulates the enhanced biosynthesis and release of LXA_4_ and LXB_4_, thereby modulating polymorphonuclear leukocyte function at local inflammatory sites [25]. The second biosynthetic pathway involves the generation of LXs by leukocytes and platelets in peripheral blood. In this pathway, ARA is initially converted to LTA_4_ by 5-LOX in leukocytes. The generated LTA_4_ is then taken up by platelets and metabolized by 12-LOX to yield LXA_4_ and LXB_4_ [26]. In inflamed tissues, platelets undergo aggregation induced by thromboxane A_2_ to facilitate hemostasis. Additionally, platelets may actively contribute to the resolution of inflammation by ingesting LTA_4_ released from recruited leukocytes and metabolizing it to anti-inflammatory LXs.

Considering the reported induction of 15-LOX in PGD_2_-exposed leukocytes [27], leukocytes accumulating at PG-abundant inflammatory sites may express some levels of 15-LOX. Consequently, lipid metabolism in leukocytes at the site of inflammation shifts from the 5-LOX-mediated production of pro-inflammatory lipid mediators, such as LTB_4_ and cysteinyl leukotrienes, to the 15-LOX-mediated production of the inflammation-resolving mediator LXA_4_ [28] (Figure 2). Additionally, aspirin, an anti-inflammatory drug, irreversibly acetylates COX-2, shifting its production from 15*S*-HETE to 15*R*-HETE [29]. Subsequently, 15*R*-HETE and 15-HpETE generated and secreted from various epithelial cells in the above-mentioned manner are taken up by leukocytes and vascular endothelial cells, and they are metabolized by endogenous 5-LOX to produce 15-epi-LXA_4_, LXA_4_, and LXB_4_ (Figure 2). The 15-epi-LXA_4_, termed aspirin-triggered LX (AT-LX), exerts anti-inflammatory effects as a lipid mediator analogous to other LXs [28,30]. Therefore, upon aspirin administration, the release of pro-inflammatory mediators is attenuated, whereas SPM production is augmented. These findings indicate that LX biosynthesis occurs via a transcellular mechanism involving the conversion of ARA to intermediate metabolites in one cell type, followed by their release and uptake by another cell [6,7,31].

Unlike the transcellular biosynthesis observed in other immune cells, mast cells generate LXA_4_ alone by the pinocytosis of 12-LOX-containing platelet microparticles [32]. Additionally, 25(OH)_2_D_3_ stimulation has been reported to induce LXA_4_ production in mast cells [33]. These findings indicate that mast cells have the capacity for monocellular biosynthesis, which differs from the transcellular biosynthesis system observed in other cell types. Furthermore, we investigated whether mast cells alone could produce LXs using mouse bone marrow-derived mast cells (BMMCs). The antigen stimulation of immunoglobulin E (IgE)-sensitized BMMCs resulted in a marked elevation in LXA_4_ and LXB_4_ levels in the culture medium, indicating the monocellular capacity of mast cells to synthesize LXs instead of transcellular mechanisms.The biosynthesis of lipoxin A_4_ and B_4_ in vivo primarily occurs through three distinct pathways. In the first pathway, arachidonic acid is converted to leukotriene A_4_ by 5-lipoxygenase, serving as a precursor for lipoxin synthesis. The second pathway involves the initial oxidation of arachidonic acid at the 12th position by 12-lipoxygenase to form 12-HpETE, which is subsequently converted to lipoxin. The third pathway involves the sequential actions of 15-lipoxygenase and 5-lipoxygenase on arachidonic acid to generate lipoxin.

### 2.2. Lipoxin Receptors and Their Expression in Mast Cells

ALX/FPR2 [34], GPR32/DRV1 [35], and BLT1 [36] have been identified as receptors for LXA_4_, whereas receptors for LXB_4_ have not yet been identified. ALX/FPR2 is a subfamily of G protein-coupled receptors (GPCRs), and it is classified as a chemoattractant receptor responding to formyl-methionyl-leucyl-phenylalanine [37]. Although many studies have reported that this receptor is coupled to Gi/o [38,39,40], its intracellular signaling may be mediated by the Gβγ-dependent activation of phospholipase, phosphatidylinositol 3-kinase, and mitogen-activated protein kinase [41]. ALX/FPR2 is highly expressed in mammalian neutrophils, dendritic cells, and microglial cells [42] and interacts with various ligands, including LXA_4_ [43], resolvin D (RvD)_1_ [44], serum amyloid A [45], Aβ42 [46], and annexin-A_1_ (lipocortin I) [47]. Conversely, aspirin-triggered 15-epi-LXA_4_ has been reported to act as an inverse agonist for the ALX/FPR2 receptor [48]. Synthetic agonists for ALX/FPR2 include the peptides WKYMVM and WKYMVm [49] and the non-peptide BML-111 [50]. Conversely, *N*-tert-butoxycarbonyl-Phe-Leu-Phe-Leu-Phe (BOC-2) [51] and Trp-Arg-Trp-Trp-Trp-NH_2_ (WRW4) [52] have been identified as antagonists.

Studies on the relationship between ALX/FPR2 and mast cells have revealed that ALX/FPR2 stimulation suppresses the compound 48/80-induced degranulation of cord blood-derived mast cells and BMMCs [53]. Additionally, BML-111, an ALX/FPR2 agonist, has been reported to inhibit the ultraviolet B-induced activation of mouse skin mast cells and subsequent skin inflammation [50]. Furthermore, pre-incubation with LX_4_, RvD_1_, or D_2_ has been reported to inhibit histamine release from human lung mast cells stimulated by FcεRI aggregation [54]. These findings support the hypothesis that the ALX/FPR2-LX axis plays a regulatory role in mast cell activation.

GPR32/DRV1 is a member of the chemoattractant pattern recognition receptor subfamily of GPCRs [55], shares similarities with ALX/FPR2, and is expressed in various cell types, including human monocytes, neutrophils [56], skin epithelial cells [57], and oral epithelial cells [58]. Although this receptor is expressed in humans, its expression in rodents has not been reported [59]. In addition to being a receptor for LXA_4_, GPR32/DRV1 binds Rvs of the D and E series [57]. Although NCGC00135472 has been identified as a synthetic agonist for GPR32/DRV1 [60], the corresponding antagonist has not been reported. GPR32/DRV1 expression has been observed in various cell types. However, no evidence of its expression in mast cells has been reported. The functional role of GPR32/DRV1 in mast cells remains unclear.

BLT1, initially identified as a receptor for LTB_4_, has been reported to bind to LXA_4_ [37]. Furthermore, this study showed that LXA_4_ inhibits LTB_4_-induced LL-37 production through BLT1. However, no study has indicated that BLT1 functions as a receptor for LXA_4_. Incidentally, mast cells express BLT1, and stimulation with LTB_4_, a BLT1 ligand, induces chemotaxis, degranulation through protein kinase B (Akt) and extracellular signal-regulated kinase (Erk) phosphorylation [61,62], and enhanced IL-8 production through NF-κB activation [63]. Additionally, the aryl hydrocarbon receptor has been suggested as an alternative receptor for LXs. However, evidence suggests that this receptor is involved in the LX-induced upregulation of SOCS2 [64].

### 2.3. The Regulatory Role of Lipoxins in Mast Cell Function

Mast cells have been reported to express at least 5-LOX [65], 12-LOX [66], and 15-LOX [67], which metabolize ARA into 5-HpETE, 12-HpETE, and 15-HpETE, respectively [68,69]. They have the potential to biosynthesize LXs through various pathways. 5-HpETE can be converted to LXs via LTA_4_, 15-HpETE can be converted to LXs via 15-OH-LTA_4_, and 12-HpETE can be directly converted to LXs (Figure 2). Additionally, when 12-LOX in platelet-derived microparticles, which are released as membrane fragments during platelet degranulation, is phagocytosed by mast cells, they begin to produce LXA_4_ [32]. Furthermore, mast cell stimulation with 1,25(OH)_2_D_3_ enhances the production of LXA_4_ and 15*S*-HETE [33]. Interestingly, the 1,25(OH)_2_D_3_-induced production of LXA_4_ is further augmented by the inhibition of NF-κB p65. As previously mentioned, LXs are produced through transcellular biosynthesis involving multiple cell types, such as neutrophils, platelets, and macrophages. However, mast cells appear to produce LXA_4_ and LXB_4_ independently upon antigen stimulation. Furthermore, IgE-sensitized BMMCs initiate the production of LXA_4_ and LXB_4_ upon FcεRI cross-linking by an antigen. Notably, LXB_4_ production is approximately 10 times higher than that of LXA_4_. Considering the limited number of studies on cells producing LXB_4_, mast cells may not only be the primary source of LXB_4_ in vivo but also represent one of the few cells capable of independently producing LXs.

In vitro studies investigating the effects of LXs on mast cells using BMMCs have shown that IgE-mediated degranulation is significantly inhibited by LXA_4_ and LXB_4_. Furthermore, it has been reported that the treatment of BMMCs with 1 μM LXA_4_ suppresses tumor necrosis factor-alpha (TNF-α) release by approximately 10%, whereas a similar concentration of LXB_4_ leads to a 40% reduction [70]. A previous study using rat peritoneal mast cells and RBL-2H3 cells showed that pretreatment with LXA_4_ or LXB_4_, followed by stimulation with compound 48/80, markedly inhibited degranulation compared to untreated controls [71]. Additionally, the LXA_4_ treatment of mast cells has been reported to decrease LTC_4_ release and sPLA_2_ activity [72]. In addition to these in vitro studies, in vivo studies using a mouse asthma model have shown that LXB_4_ suppresses mast cell degranulation [73].

Considering the expression of the LX receptor ALX/FPR2 [74] but not GPR32/DRV1 on the surface of mast cells, LXA_4_ and LXB_4_ are likely to suppress degranulation primarily through ALX/FPR2 signaling. Thus, activated mast cells not only induce local inflammation through the release of various mediators but also regulate it through an autocrine mechanism involving the production of anti-inflammatory lipid mediators, such as LXs.

## 3. Resolvins

### 3.1. Biosynthetic Pathways of Resolvins and Their Relationship with Mast Cells

In 2000, a novel anti-inflammatory lipid mediator, Rv, was reported to be generated from ω-3 polyunsaturated fatty acids through transcellular biosynthesis in the presence of a COX-2 inhibitor [75]. Subsequently, the aspirin-triggered generation of Rvs from EPA was reported in 2002 [76]. The Rv family is now subcategorized into the D series derived from DHA (Figure 3) [77], E series from EPA (Figure 4) [78], and T series from DPA (Figure 5) [79]. Given its anti-inflammatory properties, 18-HpEPE (Figure 4), an EPA-derived metabolite, has been proposed as a member of the Rv family.

Acetylated COX-2 or 15-LOX catalyzes the conversion of DHA to 17-HpDHA, which is metabolized to 17-HDHA by peroxidase. The subsequent conversion of 17-HDHA to 7-hydroperoxy-17-HDHA by 5-LOX initiates the biosynthetic pathway for RvD_1_, RvD_2_, and RvD_5_ (Figure 3). Conversely, 4-hydroperoxy-17-HDHA, which is derived from 17-HDHA, produces RvD_3_, RvD_4_, and RvD_6_ (Figure 3).

The presence of aspirin influences the stereochemical outcome of EPA metabolism. Unacetylated COX-2 catalyzes the *S*-specific peroxidation of EPA at the 18th position, whereas acetylated COX-2 promotes *R*-specific peroxidation at the same position, leading to the formation of 18*R*-HpEPE [80]. Therefore, aspirin-treated neutrophils metabolize 18*R*-HpEPE to an epoxide intermediate, which serves as a precursor for the biosynthesis of RvE_1_ and RvE_2_ (Figure 4) [81]. Conversely, the 15-LOX-catalyzed metabolism of EPA yields 15-HpEPE, which initiates the biosynthetic pathway for RvE_3_ and RvE_4_ [35].

Although it is known that DPA serves as a precursor for T-series Rvs (Figure 5) [78], the specific metabolic pathway beyond the conversion of DPA to 13*R*-HpDPA by COX-2 [82] has not been fully elucidated.

To date, only the production of RvD_1_ by mast cells has been reported [83]. However, our findings using BMMCs and tandem mass spectrometry analyses failed to demonstrate the generation of RvD_1_ or RvD_2_ upon FcεRI cross-linking.Protectins, maresins, and the D series of resolvins, collectively known as specialized proresolving mediators (SPMs), are biosynthesized from docosahexaenoic acid (DHA) via a series of enzymatic reactions. These reactions involve the sequential actions of lipoxygenases, peroxidases, and dehydratases on DHA, leading to the formation of a diverse array of SPMs.

### 3.2. Resolvin Receptors and Their Expression in Mast Cells

RvD_1_ participates in ligand–receptor interactions with ALX/FPR2 and DRV1/GPR32 [35]. RvD_2_ is a ligand for DRV2/GPR18 [84], whereas RvD_3_, similar to RvD_1_, binds to ALX/FPR2 [85] and DRV1/GPR32 [86]. Although no definitive receptor for RvD_4_ has been identified, RvD_5_ has been reported to bind to ALX/FPR2 and DRV1/GPR32, similarly to RvD_1_ and RvD_3_ [87], and to interact with GPR101, thereby initiating intracellular signaling [88]. Rvs exert their physiological effects by stimulating these receptors.

The RvD series plays a pivotal role in modulating macrophage function and differentiation. By stimulating ALX/FPR2 on macrophages, RvD_1_ inhibits CaMKII activity, thereby suppressing p38 and mitogen-activated protein kinase 2 activities and relocating 5-LOX to the cytoplasm. This leads to the inhibition of LTB_4_ production and the enhancement of LXA_4_ production [89]. The stimulation of DRV1/GPR32 on monocytes with RvD_1_ induces differentiation into M_2_ macrophages [90]. Conversely, DRV1/GPR32 knockdown suppresses RvD_1_-induced M_2_ polarization [91]. These findings indicate that RvD_1_ regulates macrophage differentiation through DRV1/GPR32. Conversely, the stimulation of DRV1/GPR32 by the RvD series inhibits the production of pro-inflammatory cytokines from macrophages and neutrophils [92], whereas the stimulation of DRV2/GPR18 by RvD_2_ increases cAMP levels [93] and sequentially induces the phosphorylation of Akt, Erk, and STAT1, 3, and 4 [83]. These mechanisms augment macrophage phagocytic activity. RvD_1_, RvD_4_, and RvD_5_ have been reported to bind to specific intracellular receptors in macrophages and modulate the function of the PGE_2_ receptor EP_4_. This interaction results in the suppression of the PGE_2_-mediated inhibition of phagocytosis and TNF-α production, thereby leading to altered macrophage function compared to conditions without Rvs [94]. These findings indicate that RvDs play a regulatory role in macrophage function.

Besides their effects on macrophages, the Rvs of the D series exhibit various physiological effects on other cell types. RvD_5_ has been reported to dose-dependently inhibit the LPS-induced production of pro-inflammatory cytokines, such as IL-1β, TNF-α, and IL-6 [95], and suppress PGE_2_-mediated coronary vasoconstriction [96]. RvD_1_ and RvD_2_ have also been reported to suppress the differentiation of naïve T cells into Th1 and Th17 cells through GPR32 and ALX/FPR2, respectively [97]. Additionally, Rvs have been suggested to regulate microglial function [80]. These findings indicate that the RvD series plays a regulatory role in the function of not only macrophages but also microglia and T cells through the interaction of these receptors. In vivo studies using a mouse peritonitis model have shown that RvD_1_ inhibits neovascularization in retinopathy [98] and suppresses neutrophil infiltration [99]. Furthermore, RvD_3_ has been reported to inhibit inflammatory pain and hyperalgesia in inflammatory pain models [89]. Unfortunately, whether these in vivo effects are mediated by receptor interactions remains unclear.

RvE_1_ has been reported to inhibit formalin-induced pain and the concomitant production of TNF-α [100]. ERV1/ChemR23, which is thought to be a receptor for RvE_1_, and the Gαi protein coupled to this protein may be involved in part of this action. The RvE_1_ stimulation of ERV1/ChemR23 has been reported to enhance the phosphorylation of Erk1/2 and Akt, thereby increasing phagocytosis, decreasing NF-κB and IL-12p40 activation, and inhibiting platelet aggregation [101,102]. Moreover, the RvE_1_ stimulation of ChemR23 on M_1_ macrophages promotes IL-10 production and subsequent M_2_ macrophage repolarization [103].

RvT, which was first identified in the exudate of infected mouse wounds, reduces the release of neutrophil extracellular traps from neutrophils while enhancing macrophage phagocytosis [104]. Additionally, this class of Rvs may modulate the biological actions of statins [85]. However, its receptor remains unknown.

Among the Rv receptors, only ALX/FPR2 has been reported to be expressed in mast cells [50,53,74]. Currently, the stimulation of ALX/FPR2, which is expressed in mast cells, is the only reported effect that inhibits the migration and activation of mast cells [54,105]. ALX/FPR2 inhibits the production of pro-inflammatory cytokines, such as IL-1β, IL-6, and TNF-α, in inflammatory cells, such as macrophages, by suppressing NF-κB activation [41]. Furthermore, ALX/FPR2 has been reported to modulate intracellular calcium concentrations, thereby regulating the activities of cAMP-dependent protein kinase, phospholipase C, and Erk phosphorylation [41]. Mast cells undergo degranulation in response to elevated intracellular calcium levels, whereas NF-κB activation stimulates pro-inflammatory cytokine production. Based on these findings, it can be inferred that the Rv family inhibits histamine release and cytokine production through ALX/FPR2, thereby negatively regulating mast cell function at the inflammatory site.

Resolvin E_3_ and E_4_, members of the E-series resolvins, are biosynthesized from eicosapentaenoic acid (EPA) through a metabolic pathway involving initial oxidation by 15-lipoxygenase, followed by further metabolism by 5-lipoxygenase and other enzymes. In contrast, resolvin E_1_ and E_2_ are derived from EPA via a pathway involving aspirin-triggered cyclooxygenase, followed by 5-lipoxygenase.

### 3.3. The Regulatory Role of Resolvins in Mast Cell Function

RvD, RvE, and RvT are biosynthesized from DHA, EPA, and DPA, respectively (Figure 3, Figure 4 and Figure 5), through the actions of COX-2, acetylated COX-2, 12-LOX, and 15-LOX. Considering that mast cells express low levels of 12-LOX, 15-LOX, and 5-LOX [67], the possibility that mast cells can produce Rvs cannot be ruled out. IgE-sensitized human mast cells have been shown to produce RvD_1_ in response to antigen challenge [83]. Additionally, various human-derived mast cells, including cord blood-derived mast cells, LAD-2 cells, foreskin-derived mast cells, nasal polyp mast cells, and BMMCs, have been reported to produce and release RvD_1_ into the supernatant upon IgE cross-linking in the presence of DHA. In this study [83], RvD_1_ levels were approximately 4 ng/mL in humans and 40 pg/mL in BMMCs. Furthermore, intracellular concentrations of RvD_1_ and E_1_ in RBL-2H3 cells have been reported to increase in an insulin-dose-dependent manner (0.1–10,000 ng/mL) after 6 days of insulin treatment [106].

The effects of Rvs on mast cells through their receptors are described in Section 3.2. Whether the observed effects are mediated by the aforementioned Rv receptors remains unclear. However, RvD has been shown to modulate the production and release of various inflammatory mediators by regulating miRNA expression. The inhibition of neutrophil infiltration has been shown to involve multiple mechanisms, such as increased miR-146b expression by RvD_1_, which leads to the inhibition of NF-κB nuclear translocation; increased miR-219 expression, which leads to the inhibition of 5-LOX and LT production; and increased miR-21 expression [107]. Additionally, RvD_1_ has been reported to upregulate TGF-β1 and IL-10 expression while downregulating IL-17 expression by modulating miRNA 30e-5p expression [104]. Among them, miR-219 is the only microRNA expressed in mast cells [108].

Based on these reports, the Rv family not only modulates the distinct functions of various inflammatory cells via receptors, such as DRV1/GPR32, DRV2/GPR18, ALX/FPR2, GPR101, and ERV1/ChemR23, but also is likely to control local inflammation by upregulating miR-219. Given that cathelicidin, a ligand for FPR2, increases the surface expression of Toll-like receptors on mast cells via FPR2 [109], Rvs and the aforementioned LXs can modulate the surface expression of Toll-like receptors on mast cells through FPR2.

## 4. Pritectin D and Maresins

### 4.1. Biosynthetic Pathways of Protectin D and Maresins and Their Relationships with Mast Cells

PD was first identified as a defensive mediator released from APRE-19 cells, which is a human retinal pigment epithelial cell line, in response to stimulation with A-23187 or IL-1β [110]. Besides APRE-19 cells, PD is produced in hepatocytes [111], peripheral blood mononuclear cells [112], and neurons [113]. The biosynthesis of PD_1_ involves the conversion of DHA to 17-*S*-HpDHA by 15-LOX, followed by the formation of an epoxide intermediate (Figure 3) [114,115]. COX-2 acetylation by aspirin leads to the production of the optically active 17-*R*-HpDHA from DHA, but the sequential formation of PD_1_ is not observed (Figure 3) [116].

MaR is an anti-inflammatory lipid mediator released from the macrophages of mice with peritonitis [117]. Following the identification of MaR_1_, 13*R*,14*S*-dihydroxy-docosahexaenoic acid (MaR_2_), which is a structurally related compound produced by macrophages, was identified [118]. In addition to macrophages, neutrophils have been reported to produce MaRs [119]. In M_2_ macrophages, DHA is converted to 14-HpDHA by intracellular 12-LOX or 15-LOX and subsequently metabolized into MaR_1_ and MaR_2_ (Figure 3) [120]. The 12-LOX expression has been reported to positively correlate with the differentiation of monocytes into macrophages and dendritic cells [121], and this correlation supports the hypothesis that MaRs are endogenous regulators of monocyte differentiation. Despite the expression of 12/15-LOX in mast cells, no study has reported the production of PD and MaRs by these cells. Our investigation using IgE-sensitized BMMCs revealed negligible PD_1_ production but significant MaR_1_ production in response to antigen challenge. The suppressive effect of MaRs on ultraviolet B-induced skin inflammation [122] and the activation of mast cells, which are abundant in the skin, by ultraviolet irradiation indicate that mast cells may regulate the severity of local skin inflammation by producing MaRs.The resolvin T series is biosynthesized from docosahexaenoic acid (DHA), but the detailed metabolic pathways remain largely unknown. Additionally, as shown in Figure 5, various specialized proresolving mediators (SPMs) have been reported to be produced from DHA.

### 4.2. Protectin D and Maresin Receptors and Their Expression in Mast Cells

PD has been reported to bind to GPR37, which is a receptor enriched in the brain and implicated in neurodegenerative diseases, such as Parkinson’s disease [123]. Furthermore, GPR37 activation by PD1 has been reported to elicit an increase in intracellular Ca^2+^ levels and enhance phagocytic activity in macrophages [124].

MaRs bind to retinoic acid-related orphan receptor alpha (RORα) [125], and the stimulation of RORα by MaR_1_ induces the differentiation of monocytes into M_2_ macrophages [124]. These findings indicate the potential role of MaRs in the anti-inflammatory properties of M_2_ macrophages. Additionally, by binding to LGR6, which is a type of GPCR, MaR_1_ has been reported to inhibit the phosphorylation of cAMP-responsive element-binding protein and Erk1/2 [126], thereby enhancing phagocytosis [127] and suppressing IL-13 production from lymphocytes [128].

### 4.3. The Regulatory Role of Protectin D and Maresins in Mast Cell Function

Although PD is biosynthesized from DHA to 17-HpDHA by 15-LOX, MaRs are generated from DHA to 14-HpDHA by 12-LOX (Figure 3). Considering the expression of 12-LOX and 15-LOX in mast cells, the production of both SPMs by these cells can be anticipated. However, no previous studies have reported the production of PD or MaRs by mast cells. Therefore, we quantified PD and MaRs in the supernatants of activated BMMCs using liquid chromatography with tandem mass spectrometry. Although PD levels remained unchanged, a significant increase in MaR_1_ production was observed.

## 5. Conclusions

This review discussed the relationship between various SPMs and mast cells, focusing on the production of SPMs by mast cells and the effects of SPMs on mast cell function. Phylogenetic analysis indicates that mast cells are present in all animals and are typically positioned at the body’s interfaces with the external environment. These findings support the hypothesis that mast cells are one of the earliest immune cell types to defend against external pathogens at the body’s boundaries [129]. This cell type has gradually shifted its role to function not only as a cell of the innate immune system but also as a component of the adaptive immune system, thereby regulating local immune responses [130]. Mast cells equipped with 5-LOX, 12-LOX, and 15-LOX can synthesize various SPMs. Although these cells are known to produce pro-inflammatory lipid mediators, such as LTB_4_ and cysteinyl LTs, upon varied stimulation, they can independently generate anti-inflammatory SPMs, such as LXA_4_, LXB_4_, PD_1_, and MaR_1_. Mast cells can modulate local inflammation by producing pro-inflammatory and anti-inflammatory lipid mediators, thereby fine-tuning the balance between the host’s defense against pathogens and tissue damage and repair. We described the biosynthetic pathways of some SPMs in mast cells and summarized the current knowledge regarding the expression of SPM receptors in mast cells. Although a large number of detailed studies have elucidated the role of mast cells in the initiation and exacerbation of inflammatory responses, the biosynthetic pathways of many SPMs and the roles of receptors and SPMs in mast cells have not been thoroughly investigated. Future investigations into the synthesis of SPMs and the receptor-mediated regulatory mechanisms of mast cells will likely provide new insights into the fundamental role of mast cells in host defense.

## Figures and Tables

**Figure 1 ijms-26-01491-f001:**
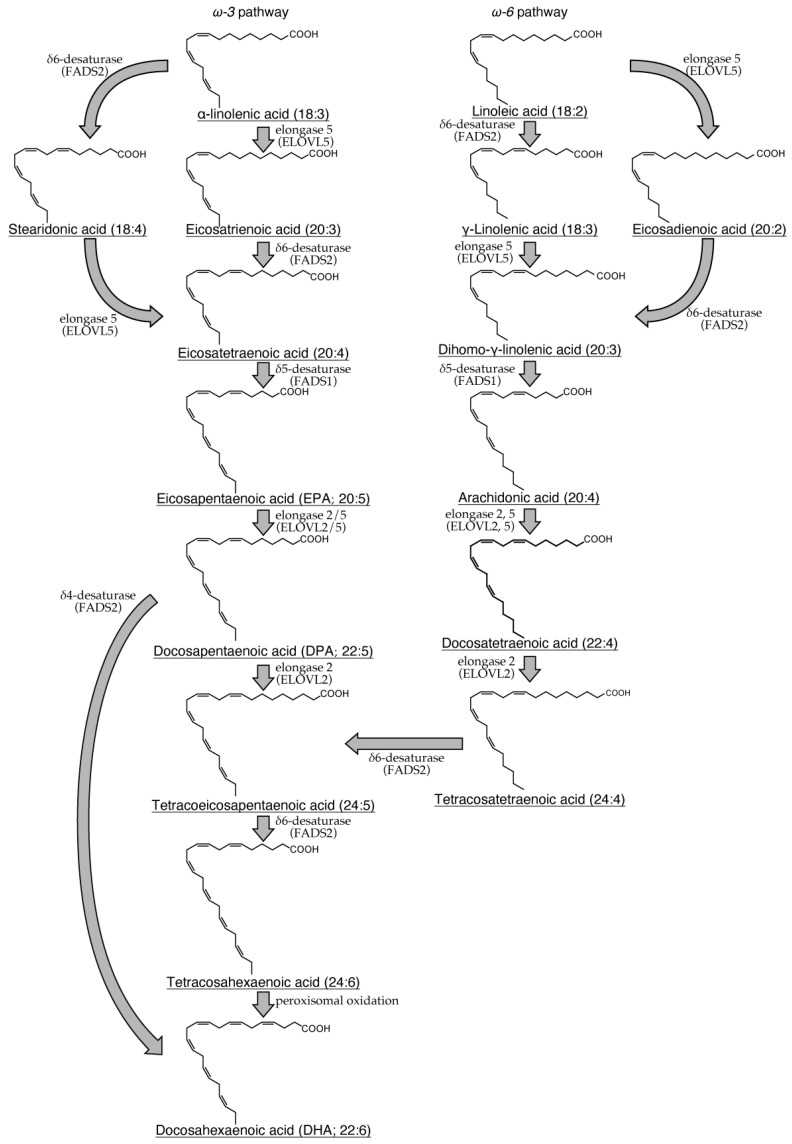
Metabolic pathways of α-linolenic acid and linoleic acid and their products in humans.

**Figure 2 ijms-26-01491-f002:**
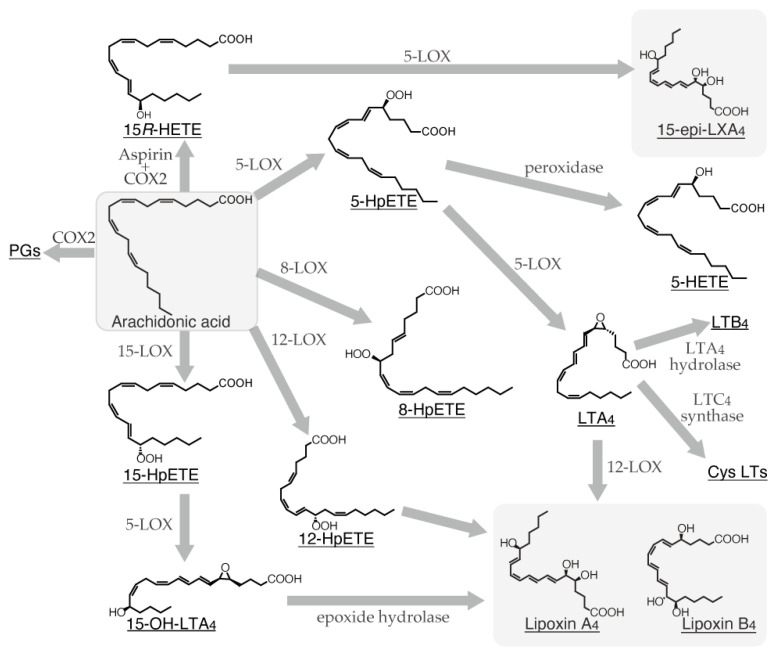
Biosynthetic pathway of lipoxin family from arachidonic acid.

**Figure 3 ijms-26-01491-f003:**
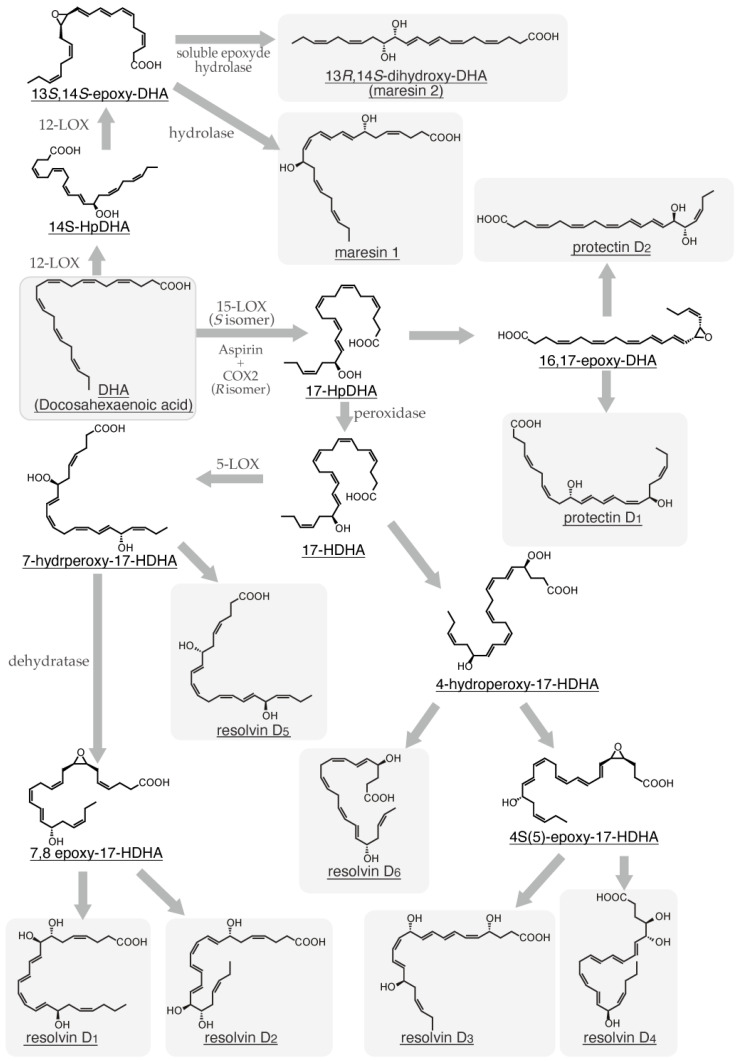
Biosynthetic pathway of protectin, maresin, and D-type resolvin families from docosahexaenoic acid.

**Figure 4 ijms-26-01491-f004:**
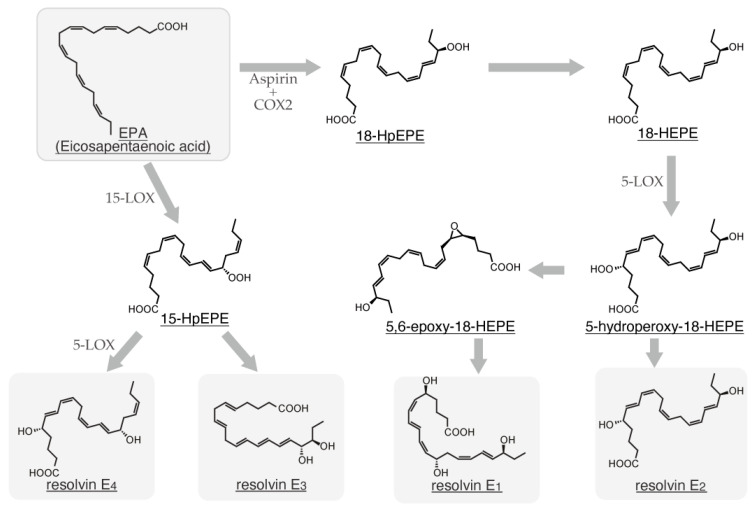
Biosynthetic pathway of E-type resolvin family from eicosapentaenoic acid.

**Figure 5 ijms-26-01491-f005:**
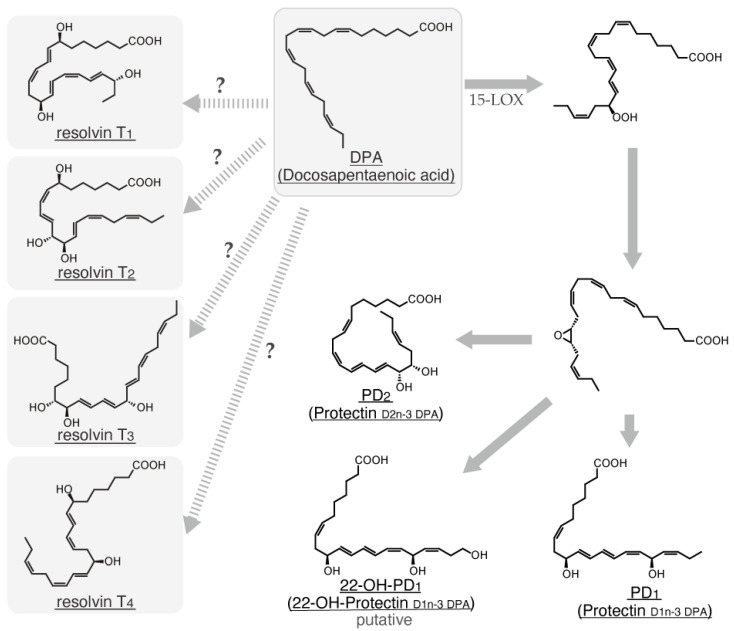
Biosynthetic pathway of T-type resolvin family and other SPMs from docosahexaenoic acid. The “?” symbol means that the biosynthetic pathway is unclear.

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
