# Peer review of "The Role of Endogenous Specialized Proresolving Mediators in Mast Cells and Their Involvement in Inflammation and Resolution"

_ijms, 2025, doi:10.3390/ijms26041491_

Round 1

Reviewer 1 Report

Comments and Suggestions for Authors

This submission reviewed the functions of specialized pro-resolving mediators in mast cells and the related applications. The topic of this review was of high significance and high quality. The length was suitable. I recommend a minor revision on the following points:

1. At first by glancing at the title I thought the SPMs were artificial ones with bio-engineering applications. Actually, the authors mainly referred to the physiological ones. If possible, I recommend the authors to add some artificial ones in a separate section following the physiological ones to broaden the scientific soundness of this review.

2. The figures in this review was of high quality. If they referred the patterns from the previous investigations, I recommend the copyright statement or at least the citation. If they were absolutely original, I would appreciate the standard pattern.

3. I recommend the addition of the discussion or prospect section before the conclusion. The authors should point some future investigation orientations to improve the guiding significance of this review.

4. The language use might be further improved.

Comments on the Quality of English Language

Moderate revision needed.

Author Response

Reviewer 1

This submission reviewed the functions of specialized pro-resolving mediators in mast cells and the related applications. The topic of this review was of high significance and high quality. The length was suitable. I recommend a minor revision on the following points:

  1. At first by glancing at the title I thought the SPMs were artificial ones with bio-engineering applications. Actually, the authors mainly referred to the physiological ones. If possible, I recommend the authors to add some artificial ones in a separate section following the physiological ones to broaden the scientific soundness of this review.

Response

Thank you for your kindly pointed out about the title of the manuscript.  Specialized proresolving mediators (SPMs) are generally defined as anti-inflammatory mediators composed of endogenous polyunsaturated fatty acids. Therefore, artificial SPMs have been scarcely reported, and their effects on mast cells, to the best of our knowledge, have not been reported.

On the other hand, as it was pointed out that the title suggests artificial compounds, replaced title was considered as follows:

Lines 2-4;

“The Role of Endogenous Specialized Proresolving Mediators in Mast Cells and Their Involvement in Inflammation and Resolution”

  1. The figures in this review was of high quality. If they referred the patterns from the previous investigations, I recommend the copyright statement or at least the citation. If they were absolutely original, I would appreciate the standard pattern.

Response

Thank you for your useful advice to chemical structural formula in the figures. The chemical structure formulas we described were completely original, but they were not in the general format as reviewer pointed out. We rewrote all the structural formulas in the figures into standard pattern and regenerated all of the figures in the manuscript as follows.

Line 84; replaced new Figure 1

Line 142; replaced new Figure 2

Line 254; replaced new Figure 3

Line 322; replaced new Figure 4

Line 385; replaced new Figure 5

Thank you.

  1. I recommend the addition of the discussion or prospect section before the conclusion. The authors should point some future investigation orientations to improve the guiding significance of this review.

Response

Thank you for your useful advice. General IMJS format for review article is consist of abstract, introduction, main body, and conclusion. Therefore, addition of the following sentences regarding future investigation orientations to the last part of the conclusion was considered.

Lines 426-434;

“We described the biosynthetic pathways of some SPMs in mast cells and summarized the current knowledge regarding the expression of SPM receptors in mast cells. Although a large number of detailed research have elucidated the role of mast cells in the initiation and exacerbation of inflammatory responses by mast cells, the biosynthetic pathways of many SPMs and the receptors and their roles of SPMs in mast cells have not been thoroughly investigated. Future investigations into the synthesis of SPMs and the receptor-mediated regulatory mechanisms of mast cells will likely provide new insights into the fundamental role of mast cells in host defense.”

  1. The language use might be further improved.

Response

Thank you for your pointed out. We attached certification of English grammar check for professional commercial service.

Thank you.

Reviewer 2 Report

Comments and Suggestions for Authors

The review is well-organized and provides a comprehensive summary of the current knowledge on inflammation inducers and resolvers. It specifically explores the production of specialized proresolving mediators (SPMs) by mast cells and their impact on these cells, emphasizing their role in resolving inflammation. By examining this interaction, the review offers valuable insights into how mast cells contribute to immune homeostasis and tissue integrity.

I have few comments

  1. Avoid using short abbreviation in main heading and sub-headings in the manuscript like in Page3/18 line 98
  2. Avoid long sentences, as they can create confusion. Many sentences are lengthy and complex, making it difficult to draw clear conclusions. For example, sentences 117 to 120 are particularly confusing it's unclear whether the author is stating that the metabolic shift in leukocytes within inflammation foci promotes the production of LTB4, a pro-inflammatory mediator, or LXA4, a pro-resolving mediator120-123 acetylation of Cox-2 by Aspirin alters catalytic activity of which enzyme, that prefentially promotes conversion of ARA to 15R-HETE, is not clear.
Comments on the Quality of English Language

Avoid long complex sentences. 

Author Response

Reviewer 2

The review is well-organized and provides a comprehensive summary of the current knowledge on inflammation inducers and resolvers. It specifically explores the production of specialized proresolving mediators (SPMs) by mast cells and their impact on these cells, emphasizing their role in resolving inflammation. By examining this interaction, the review offers valuable insights into how mast cells contribute to immune homeostasis and tissue integrity.

I have few comments

  1. Avoid using short abbreviation in main heading and sub-headings in the manuscript like in Page3/18 line 98.

Response

Thank you for your useful advice. All of the main- and sub-headings in the manuscript was replaced as below.

Line 94; 2. Lipoxins

Line 95; 2.1) Biosynthetic pathways of lipoxins….

Line 151; 2.2) Lipoxin receptors….

Line 193; 2.3) The regulatory role of lipoxins….

Line 227; 3. Resolvins

Line 228; 3.1) Biosynthetic pathway of resolvins….

Line 329; 3.3) The regulatory role of resolvins….

Line 359; 4. Protectin D and maresins

Line 360; 4.1) Biosynthetic pathway of protectin D and maresins….

Line 392; 4.2) Protectin D and maresin receptors….

Line 404; 4.3) The regulatory role of protectin D and maresins….

  1. Avoid long sentences, as they can create confusion. Many sentences are lengthy and complex, making it difficult to draw clear conclusions. For example, sentences 117 to 120 are particularly confusing it’s unclear whether the author is stating that the metabolic shift in leukocytes within inflammation foci promotes the production of LTB4, a pro-inflammatory mediator, or LXA4, a pro-resolving mediator120-123 acetylation of Cox-2 by Aspirin alters catalytic activity of which enzyme, that prefentially promotes conversion of ARA to 15R-HETE, is not clear.

Response

Thank you for your pointed out. The sentences reviewers pointed out were replaced as below.

  1. about lines 117-120

In this sentence, we aim to address the following:

1) In leukocytes accumulated at the site of inflammation, arachidonic acid metabolism shifts from 5-LOX to 15-LOX.

2) As a result of 1), in leukocytes accumulated at the site of inflammation, both decreased production of LTB4 which is a pro-inflammatory mediator and increased production of LXA4 that is an anti-inflammatory mediator are observed.

Therefore, we would like to rewrite the following sentence:

Lines 118-121

“Consequently, lipid metabolism in leukocytes at the site of inflammation shifts from the 5-LOX-mediated production of proinflammatory lipid mediators, such as LTB4 and cysteinyl leukotrienes, to the 15-LOX-mediated production of the inflammation-resolving mediator LXA4.”

  1. about lines 120-123

In this sentence, we aim to address the following:

1) aspirin, which is commonly used as anti-inflammatory drug, irreversibly acetylates COX-2.

2) as a result of 1), decreased production of PGG2 via 15(S)-HETE from arachidonic acid by non-acetylated COX-2 and increased production of 15-epi-LXA4 via 15(R)-HETE from arachidonic acid by acetylated COX-2 are observed.

We would like to rewrite the following sentence:

Lines 121, 122

“Additionally, aspirin, an anti-inflammatory drug, irreversibly acetylates COX-2, shifting its production from 15S-HETE to 15R-HETE.“

Thank you.